# Mortality rates among COVID-19 patients hospitalised during the first three waves of the epidemic in Milan, Italy: A prospective observational study

**Andrea Giacomelli**[1]ᴼ*, **Anna Lisa Ridolfo**[1]ᴼ, **Laura Pezzati**[1,2], **Letizia Oreni**[1], **Giorgia Carrozzo**[1,2], **Martina Beltrami**[1,2], **Andrea Poloni**[1,2], **Beatrice Caloni**[1,2], **Samuel Lazzarin**[1,2], **Martina Colombo**[1,2], **Giacomo Pozza**[1,2], **Simone Pagano**[1,2], **Stefania Caronni**[1,2], **Chiara Fusetti**[1], **Martina Gerbi**[1], **Francesco Petri**[1], **Fabio Borgonovo**[1], **Fabiana D'Aloia**[2], **Cristina Negri**[1], **Giuliano Rizzardini**[1], **Spinello Antinori**[1,2]

**1** Infectious Diseases Department, ASST Fatebenefratelli Sacco, Ospedale Luigi Sacco, Milan, Italy,
**2** Department of Biomedical and Clinical Sciences, University of Milan, Milan, Italy

ᴼ These authors contributed equally to this work.
* andrea.giacomelli@asst-fbf-sacco.it, dott.giacomelli@gmail.com

**Data Availability Statement:** All relevant data are within the manuscript and its Supporting Information files.

## Abstract

### Introduction

This paper describes how mortality among hospitalised COVID-19 patients changed during the first three waves of the epidemic in Italy.

### Methods

This prospective cohort study used the Kaplan-Meier method to analyse the time-dependent probability of death of all of the patients admitted to a COVID-19 referral centre in Milan, Italy, during the three consecutive periods of: 21 February-31 July 2020 (first wave, W1), 1 August 2020–31 January 2021 (second wave, W2), and 1 February-30 April 2021 (third wave, W3). Cox models were used to examine the association between death and the period of admission after adjusting for age, biological sex, the time from symptom onset to admission, disease severity upon admission, obesity, and the comorbidity burden.

### Results

Of the 2,023 COVID-19 patients admitted to our hospital during the study period, 553 (27.3%) were admitted during W1, 838 (41.5%) during W2, and 632 (31.2%) during W3. The crude mortality rate during W1, W2 and W3 was respectively 21.3%, 23.7% and 15.8%. After adjusting for potential confounders, hospitalisation during W2 or W3 was independently associated with a significantly lower risk of death than hospitalisation during W1 (adjusted hazard ratios [AHRs]: 0.75, 95% confidence interval [CI] 0.59–0.95, and 0.58, 95% CI 0.44–0.77). Among the patients aged >75 years, there was no significant difference in the probability of death during the three waves (AHRs during W2 and W3 vs W1: 0.93, 95% CI 0.65–1.33, and 0.88, 95% CI 0.59–1.32), whereas those presenting with critical

**Funding:** The author(s) received no specific funding for this work.

**Competing interests:** The authors have declared that no competing interests exist.

disease during W2 and W3 were at significantly lower risk of dying than those admitted during W1 (AHRs 0.61, 95% CI 0.43–0.88, and 0.44, 95% CI 0.28–0.70).

## Conclusions

Hospitalisation during W2 and W3 was associated with a reduced risk of COVID-19 death in comparison with W1, but there was no difference in survival probability in patients aged >75 years.

## 1. Introduction

Almost two years after the start of the coronavirus disease 19 (COVID-19) pandemic caused by severe acute respiratory syndrome coronavirus-2 (SARS-CoV-2), more than 326,279,424 confirmed infections and 5,536,609 COVID-19 related deaths had been reported worldwide [1]. Although most SARS-CoV-2 infections were asymptomatic or only mildly or moderately symptomatic and did not require hospitalisation, hospitalisation rates ranged from 13 to 21 cases per 1,000 confirmed infections, and mainly depends on the age of the infected population [2, 3].

One of the concerning aspects of the pandemic is the probability of death among diagnosed subjects: initial analyses of data from nine countries (China, France, Germany, Italy, The Netherlands, South Korea, Spain, Switzerland and the United States) made in April 2020 showed wide variations in the overall case fatality rate (CFR), which ranged from 0.7% in Germany to 9.3% in Italy, with two-thirds of this variability being explained by differences in the age distribution of the cases [4]. In Italy, the CFR had increased to 14% by June 2020, mainly because of increasing age-specific case fatality [5]. Moreover, the first observational studies in China, Italy and the United States indicated mortality rates ranging from 12% to 28% among patients hospitalised with COVID-19 and going as high as 49% among those admitted to an intensive care unit (ICU), with the highest rates being observed among older patients and those with pre-existing comorbidities [6–8]. However, although the pandemic has continued in waves in many parts of the world and is still keeping health systems under pressure, some European and American studies have shown that the in-hospital mortality associated with COVID-19 has progressively declined [9–13]. This has been variously attributed to rapid changes in hospital organisation that allowed the better management of patient surges (i.e. expanded COVID-19 wards, increased supplies of ventilators and other critical equipment); earlier hospitalisation as a result of easier access to testing; and improvements in COVID-19 treatment [9–13], but it is also possible that the increasing circulation of SARS-CoV-2 and the implementation of measures to shield the most vulnerable populations contributed by changing the case-mix of hospitalised patients towards those who are younger and at lower risk of dying [14].

Since the arrival of the pandemic in late February 2020, Italy has experienced three major surges of COVID cases and deaths, and a fourth wave of contagion is currently ongoing. The cumulative number of COVID-19 deaths as of 9 January 2022 was 137,705 [15], with the highest number being recorded in Lombardy [16], the region that was the epicentre of the first outbreak in the country [16]. During the early days of the COVID-19 epidemic, a registry of all of the patients admitted to the Department of Infectious Diseases and the ICU of Milan's Luigi Sacco Hospital was established in order to provide initial information concerning the characteristics of the disease and its lethality [17–19].

The aim of this study of updated registry data is to describe how mortality changed during the different phases of the COVID epidemic in Italy and consider the potential drivers of the changes.

## 2. Materials and methods

### 2.1 Study design and setting

This prospective cohort study enrolled all of the adult COVID-19 patients admitted to the Department of Infectious Diseases and the ICU of Luigi Sacco Hospital. The characteristics of the hospital and the rapid reorganisation require in order to respond to the pandemic have been previously described [17–19].

### 2.2 Participants

The analysis considered all of the adult COVID-19 patients admitted to the hospital's Department of Infectious Diseases and ICU between 21 February (the day the first patients were hospitalised) and 30 April 2021, who were observed until the time of death or censored three months after hospital admission, whichever came first. The diagnosis of COVID-19 was confirmed by a positive real-time reverse-transcription polymerase chain reaction on a nasopharyngeal swab (NPS).

### 2.3 Data collection

The characteristics of our registry data management have been described elsewhere [17–19]. In brief, the data were extracted from the patients' clinical charts on a daily basis and stored in an *ad hoc* database. The collected data were the patients' date and place of birth, and biological sex; the date of admission, and the time between symptom onset and hospital admission; comorbidities (including diabetes, lung diseases, heart diseases, renal diseases, immune system diseases, liver diseases, and obesity, defined as a body mass index of ≥30) [20]; the burden of comorbidities (0, 1, 2, and 3+); disease severity upon hospital admission (defined as mild, moderate, severe or critical in accordance with the WHO guidelines for the management of COVID-19) [21]; the type of supportive oxygen therapy upon hospital admission and/or during the hospital stay; the drugs used to treat COVID-19 (which included hydroxychloroquine, lopinavir/ritonavir, remdesivir, tocilizumab and other immunomodulators, heparin, and steroids); and the hospitalisation outcome (death, discharge, or transfer to other facilities).

The vital status of the patients who were discharged before the censoring date was ascertained by means of telephone calls and, when appropriate, during an on-site examination by a member of our dedicated post-COVID-19 outpatient service.

For the purposes of this study, the patients were divided into three groups on the basis of the time of their hospital admission. The three consecutive periods were characterised by surges in COVID-19 cases and hospitalisations in Italy: 21 February-31 July 2020 (first wave, W1); 1 August 2020–31 January 2021 (second wave, W2); and 1 February-30 April 2021 (third wave, W3) (S1 Fig) [15–16, 22].

### 2.4 Outcomes

The main outcome of interest was death.

### 2.5 Data analysis

The descriptive statistics show proportions for categorical variables, and median values with their interquartile range (IQR) for continuous variables. The baseline demographic and

clinico-epidemiological characteristics of the patients admitted during W1, W2 and W3 were compared using the $\chi^2$ or, when necessary, Fisher's exact test in the case of categorical variables, and Wilcoxon's rank-sum test in the case of continuous variables.

The time-dependent probability of death among the patients admitted during W1, W 2 and W3 was assessed using the Kaplan-Meier method, with Dunn-Sidak adjustment for multiple comparisons. Univariable Cox proportional hazard models were used to investigate the relationship between the period of admission (W1, W2 or W3) and the outcome of interest (death). Associations were estimated using hazard ratios (HRs) with their 95% confidence intervals (CIs). Multivariate Cox proportional hazard models were used to adjust the HRs for potential confounders including age, biological sex, the time from symptom onset to admission, disease severity upon admission, the co-morbidity burden, and obesity, and the results were expressed as adjusted hazard ratios (AHRs) with their 95% CIs.

It was hypothesised that the burden of the spread of SARS-CoV-2 in the province of Milan may have influenced mortality rates in the different study periods as it was presumed that a larger number of infections in the general population would be associated with a larger number of severely or critically ill patients being hospitalised and greater difficulties in their management. Consequently, a variable in which each hospital admission date was attributed with the monthly number of SARS-CoV-2 infections recorded in the province of Milan [22] was created and included in the multivariable model as a possible confounder of the risk of death during W2 and W3 (W1 was excluded because the shortage of reagents for NPS testing and the policy of restricting testing to symptomatic patients affected the reliability of any estimate of the number of infected subjects during this period).

Furthermore, a sub-analysis of the risk of death was carried out by stratifying the patients on the basis of their age (<46 years; 46–60 years; 61–75 years; >75 years) and disease severity upon hospital admission (the WHO categories of mild/moderate; severe; critical). We also explored changes in survival probability across the waves in subgroups of patients stratified by multiple variables including sex, age, clinical conditions at hospital admission, and comorbidity burden.

The study was approved by our *Comitato Etico Interaziendale Area 1* (Protocol No. 16088). Informed consent was waived in the case of patients undergoing mechanical ventilation upon admission.

## 3. Results

Of the 2,023 COVID-19 patients admitted to our centre during the study period, 553 (27.3%) were admitted during W1, 838 (41.5%) during W2, and 632 (31.2%) during W3.

### 3.1 Admission characteristics

Table 1 shows the distribution of the patients' characteristics upon admission as a whole and by wave.

Most of the patients were males (64.3%), and there was no between-wave difference in the distribution of their biological sex. The patients admitted during W1 were younger than those admitted during W2 or W3 (60.8 years [IQR 49.3–72.6] *vs* 68.5 years [IQR 56–80.1] and 66.4 years [IQR 56.9–75.8]), and patients aged >75 years were more represented in W2 than W1 or W3 (36.3% *vs* 20.6% and 26.6%). The patients admitted during W1 were also more frequently free of comorbidities than those admitted during W2 or W3 (31.3% *vs* 23.3% and 25.9%). The median time from symptom onset to hospitalisation was shorter among the patients admitted during W2 than among those admitted during W1 or W3 (6 days [IQR 3–9] *vs* 7 days [IQR 4–10] and 8 days [IQR 5–11]). The number of patients requiring immediate oxygen support

**Table 1. Characteristics of the study population as a whole and by period of hospital admission.**

| | | Wave | | |
|---|---|---|---|---|
| | Overall n = 2023 | 1 (21 Feb—31 Jul 2020) n = 553 | 2 (1 Aug 2020–31 Jan 2021) n = 838 | 3 (1 Feb– 30 Apr 2021) n = 632 |
| **Biological sex, n (%)** | | | | |
| Female | 722 (35.7) | 185 (33.5) | 300 (35.8) | 237 (37.5) |
| Male | 1301 (64.3) | 368 (66.5) | 538 (64.2) | 395 (62.5) |
| **Median age, years [IQR]** | 65.7 [54.8, 76.9] | 60.8 [49.3, 72.6] | 68.5 [56, 80.1] | 66.4 [56.9, 75.8] |
| **Age, n (%)** | | | | |
| <46 | 225 (11.1) | 103 (18.6) | 78 (9.3) | 44 (7.0) |
| 46–60 | 540 (26.7) | 161 (29.1) | 205 (24.5) | 174 (27.5) |
| 61–75 | 672 (33.2) | 175 (31.6) | 251 (30.0) | 246 (38.9) |
| >75 | 586 (29.0) | 114 (20.6) | 304 (36.3) | 168 (26.6) |
| **Italian nationality, n (%)** | 1633 (80.7) | 446 (80.7) | 648 (77.3) | 539 (85.3) |
| **Comorbidities, n (%)** | | | | |
| Obesity‡, | 488 (24.1) | 98 (17.7) | 195 (23.3) | 195 (30.9) |
| Diabetes | 315 (15.6) | 62 (11.2) | 137 (16.3) | 116 (18.4) |
| Lung disease | 301 (14.9) | 86 (15.6) | 144 (17.2) | 71 (11.2) |
| Heart disease | 1124 (55.6) | 268 (48.5) | 487 (58.1) | 369 (58.4) |
| Renal disease | 170 (8.4) | 45 (8.1) | 86 (10.3) | 39 (6.2) |
| Oncological disease | 252 (12.5) | 55 (9.9) | 126 (15.0) | 71 (11.2) |
| Immune system disease | 149 (7.4) | 43 (7.8) | 67 (8.0) | 39 (6.2) |
| Liver disease | 71 (3.5) | 12 (2.2) | 39 (4.7) | 20 (3.2) |
| **No. of comorbidities, n (%)** | | | | |
| 0 | 532 (26.3) | 173 (31.3) | 195 (23.3) | 164 (25.9) |
| 1 | 661 (32.7) | 183 (33.1) | 261 (31.1) | 217 (34.3) |
| 2 | 524 (25.9) | 128 (23.1) | 224 (26.7) | 172 (27.2) |
| ≥3 | 306 (15.1) | 69 (12.5) | 158 (18.9) | 79 (12.5) |
| **Median time from symptom onset, [IQR]** | 7 [4, 10] | 7 [4, 10] | 6 [3, 9] | 8 [5, 11] |
| **O$_2$ therapy upon admission, n (%)** | | | | |
| None | 501 (24.8) | 164 (29.7) | 243 (29.0) | 94 (14.9) |
| Nasal cannula | 511 (25.3) | 126 (22.8) | 181 (21.6) | 204 (32.3) |
| Venturi | 440 (21.7) | 88 (15.9) | 178 (21.2) | 174 (27.5) |
| Reservoir | 136 (6.7) | 36 (6.5) | 51 (6.1) | 49 (7.8) |
| C-PAP | 369 (18.2) | 113 (20.4) | 158 (18.9) | 98 (15.5) |
| Mechanical ventilation | 66 (3.3) | 26 (4.7) | 27 (3.2) | 13 (2.1) |
| **Disease severity upon admission†, n (%)** | | | | |
| Mild | 167 (8.3) | 49 (8.9) | 89 (10.6) | 29 (4.6) |
| Moderate | 852 (42.1) | 241 (43.6) | 338 (40.3) | 273 (43.2) |
| Severe | 569 (28.1) | 124 (22.4) | 226 (27.0) | 219 (34.7) |
| Critical | 435 (21.5) | 139 (25.1) | 185 (22.1) | 111 (17.6) |

Abbreviations: n: number; IQR: interquartile range.

‡Defined as a body mass index of ≥30 [20]

†According to the WHO classification [21]

was lower during W3 than during W1 or W2 (14.9% *vs* 29.7% and 29.0%), but the overall proportion of patients presenting with critical disease was higher during W1 than during W2 or W3 (25.1% *vs* 22.1% *vs* 17.6%).

## 3.2 Oxygen support, treatments, and outcomes

Table 2 shows the type of oxygen support and pharmacological intervention required during hospitalisation.

Advanced respiratory support with C-PAP was required by 36.9% of patients, and invasive mechanical ventilation by 15.6%, and there was no between-wave difference in the proportion of patients who underwent either. Most of the patients (84.5%) received some oxygen support; the percentage not requiring it was lower during W3 than during W1 or W2 (4.4% *vs* 21.2% and 20.2%).

On the basis of the evolving evidence provided by clinical trials during the study period, the use of remdesivir, heparin and steroids progressively increased from 15.7%, 36.5% and 11.6% during W1 to 25.8%, 92.4% and 89.9% during W3.

Overall, the median duration of hospitalisation was 13 days (IQR 8–23), but it was longer during W2 (15 days, IQR: 9–25) than during W1 or W3 (12 days, IQR: 8–22 in both periods).

Four hundred and seventeen of the study patients died (20.6%). The crude mortality rate during W1, W2 and W3 was respectively 21.3%, 23.7% and 15.8%.

## 3.3 Survival analysis and the factors associated with the risk of COVID-19-related death

Fig 1 shows the time-dependent probability of survival during the different waves of the epidemic.

The probability of death was lower among the patients admitted during the third wave of the epidemic than during the first or second wave. The probability of being alive 30 days after

**Table 2. Characteristics of oxygen support, pharmacological treatments, and outcomes.**

| | | Wave | | |
|---|---|---|---|---|
| | Overall<br>n = 2023 | 1 (21 Feb– 31 Jul 2020)<br>n = 553 | 2 (1 Aug 2020–31 Jan 2021)<br>n = 838 | 3 (1 Feb– 30 Apr 2021)<br>n = 632 |
| **Advanced respiratory support** | | | | |
| C-PCP, n (%) | 747 (36.9) | 202 (36.5) | 303 (36.2) | 242 (38.3) |
| Invasive ventilation, n (%) | 316 (15.6) | 88 (15.9) | 144 (17.2) | 84 (13.3) |
| **Maximum level of $O_2$ support, n (%)** | | | | |
| No oxygen support | 314 (15.5) | 117 (21.2) | 169 (20.2) | 28 (4.4) |
| Nasal cannula | 391 (19.3) | 94 (17.0) | 140 (16.7) | 157 (24.8) |
| Venturi | 369 (18.2) | 76 (13.7) | 142 (16.9) | 151 (23.9) |
| Reservoir | 139 (6.9) | 39 (7.1) | 59 (7.0) | 41 (6.5) |
| C-PAP | 494 (24.4) | 139 (25.1) | 184 (22.0) | 171 (27.1) |
| Invasive ventilation | 316 (15.6) | 88 (15.9) | 144 (17.2) | 84 (13.3) |
| **Pharmacological treatment, n (%)** | | | | |
| Hydroxychloroquine | 418 (20.7) | 417 (75.4) | 0 (0.0) | 1 (0.2) |
| Lopinavir/ritonavir | 326 (16.1) | 326 (59.0) | 0 (0.0) | 0 (0.0) |
| Remdesivir | 391 (19.3) | 87 (15.7) | 141 (16.8) | 163 (25.8) |
| Tocilizumab | 123 (6.1) | 116 (21.0) | 1 (0.1) | 6 (0.9) |
| Heparin | 1517 (75.0) | 202 (36.5) | 731 (87.2) | 584 (92.4) |
| Steroids | 1281 (63.3) | 64 (11.6) | 649 (77.4) | 568 (89.9) |
| **Median number of days of hospitalisation [IQR]** | 13 [8, 23] | 12 [8, 22] | 15 [9, 25] | 12 [8, 22] |
| **Deaths, n (%)** | 417 (20.6) | 118 (21.3) | 199 (23.7) | 100 (15.8) |

Abbreviations: n: number; IQR: interquartile range.

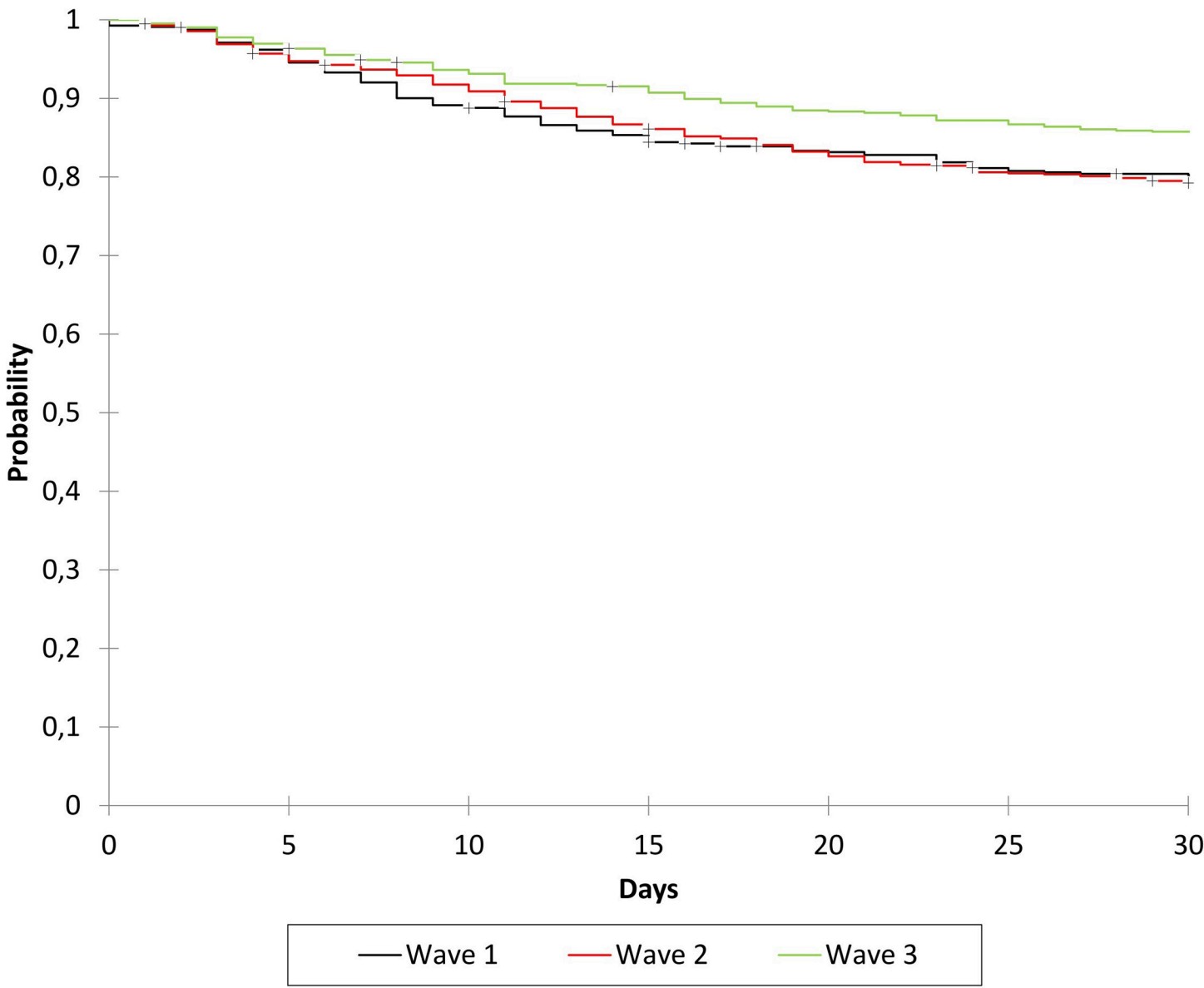

**Fig 1. Time-dependent survival probability by period of hospital admission: Wave 1 (February-July 2020), wave 2 (August 2020-January 2021), and wave 3 (February-April 2021).**

hospital admission was 80.2% (95% CI 76.9–83.6%) during W1, 79.3% (95% CI 76.5–82%) during W2, and 85.8% (95% CI 83–88.5%) during W3.

Table 3 shows the results of the uni- and multivariable analyses of the effect of the period of hospital admission on the risk of COVID-19-related death. Being hospitalised with COVID-19 during W2 or W3 was independently associated with a significantly lower risk of death in comparison with being hospitalised during W1 (AHR 0.75, 95% CI 0.59–0.95, and AHR 0.58, 95% CI 0.44–0.77).

The multivariable analyses also confirmed that an older age (AHR 2.18, 95% CI 1.99–2.39 per each additional 10 years), male sex (AHR 1.36, 95% CI 1.09–1.68), obesity (AHR 1.66, 95% CI 1.32–2.07), and disease severity (AHR 2.20 (95% CI 1.70–2.83 for severe *vs* mild/moderate

**Table 3. Cox model of the effect of the period of hospital admission on the risk of COVID-19-related death.**

|  | HR | HR 95% CI | AHR | AHR 95% CI |
|---|---|---|---|---|
| **Age (per each additional 10 years)** | 1.80 | 1.67–1.94 | 2.18 | 1.99–2.39 |
| **Time from symptom onset (per each additional day)** | 0.99 | 0.98–1.01 | 0.99 | 0.98–1.01 |
| **SARS-CoV-2 epidemic wave 2 *vs* 1** | 1.11 | 0.88–1.39 | 0.75 | 0.59–0.95 |
| **SARS-CoV-2 epidemic wave 3 *vs* 1** | 0.71 | 0.55–0.93 | 0.58 | 0.44–0.77 |
| **Males *vs* females** | 1.01 | 0.90–1.34 | 1.36 | 1.09–1.68 |
| **Obesity: yes *vs* no** | 1.22 | 0.99–1.52 | 1.66 | 1.32–2.07 |
| **No. of comorbidities ≥3 *vs* <3** | 1.58 | 1.25–1.99 | 1.00 | 0.79–1.28 |
| **Disease severity: severe *vs* mild/moderate** | 2.19 | 1.71–2.80 | 2.20 | 1.70–2.83 |
| **Disease severity: critical *vs* mild/moderate** | 3.79 | 2.99–4.80 | 5.03 | 3.90–6.49 |

Abbreviations: HR: hazard ratio; AHR: adjusted hazard ratio; CI: confidence interval.

disease, and 5.03, 95% CI 3.90–6.49 for critical *vs* mild/moderate disease) were all independently associated with a higher risk of COVID-19-related death.

A sensitivity analysis of the risk of death restricted to patients admitted during W2 and W3 showed a significantly lower hazard ratio among the latter (HR 0.64, 95% CI 0.51–0.82). However, this was not confirmed when the analysis was further adjusted for the monthly number of cases diagnosed in the province of Milan during the two epidemic waves (AHR W3 *vs* W2 = 0.85, 95% CI 0.65–1.11) (S1 Table).

### 3.4 Survival analysis by age strata and clinical condition at admission

Fig 2 shows the survival curves relating to the three epidemic waves by age strata. There was no difference between the survival curves of the patients aged >75 years during the three periods and, after adjusting for confounders, no significant difference in the risk of death between the patients admitted during W2 or W3 and those admitted during W1 was observed (AHRs 0.93, 95% CI 0.65–1.33, and 0.88, 95% CI 0.59–1.32).

However, the survival probability of the patients aged 46–60 years was different between waves: after adjusting for potential confounders, being admitted during W2 and W3 was independently associated with a significantly reduced risk of death (AHRs of 0.42 [95% CI 0.19–0.94] and 0.17 [95% CI 0.05–0.61]). The survival probability of the patients aged 61–75 years was also different between waves: the adjusted multivariable model showed that the risk of dying was significantly lower during W3 than during W1 (AHR 0.51 [95% CI 0.33–0.78]).

Fig 3 shows the survival curves by disease severity upon hospital admission during the different study periods.

There was no difference in survival between the patients admitted with severe disease during the three periods: after adjusting for confounders, the estimated risk of death of the patients admitted during W2 or W3 was not significantly different from that of the patients admitted during W1 (AHRs of 0.95 [95% CI 0.59–1.54] and 0.85 [95% CI 0.51–1.40]). However, the survival probability of the patients admitted with mild/moderate disease was different between waves: after adjusting for potential confounders, the patients admitted during W3 were at significantly lower risk of dying than those admitted during W1 (AHR 0.57 [95% CI 0.33–0.97]). Finally, the survival probability of the patients presenting with critical disease was also different between waves: after adjusting for potential confounders, the patients with critical disease admitted during W2 or W3 were at significantly lower risk of dying than those admitted during W1 (AHRs of 0.61 [95% CI 0.43–0.88] and 0.44 [95% CI 0.28–0.70]).

S2 Fig shows the results of the survival analysis by period of admission in subgroups of patients stratified by multiple characteristics. There was an overall improvement in survival in

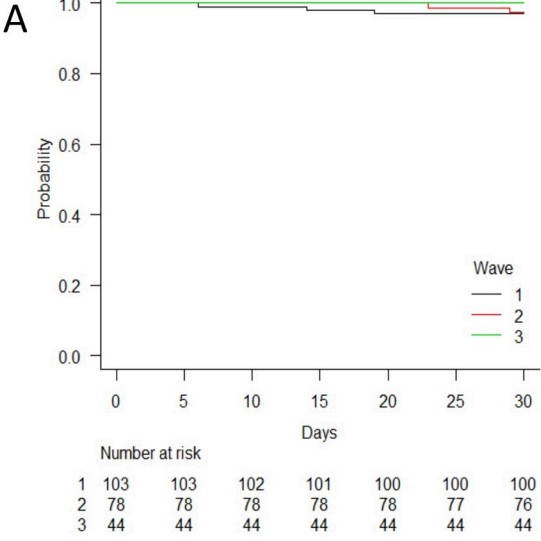

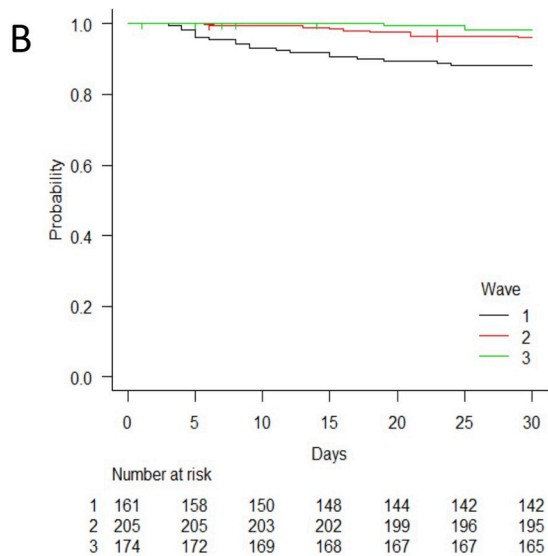

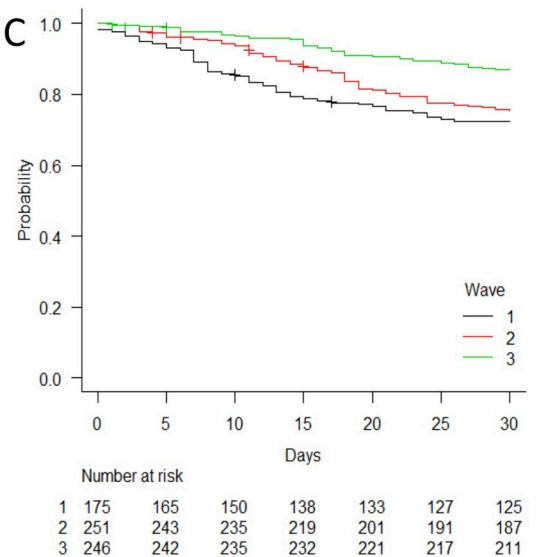

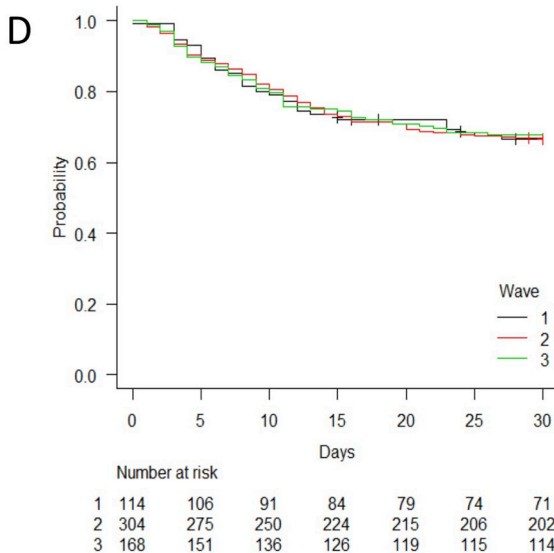

**Fig 2. Time-dependent survival probability by period of hospital admission (wave 1 [February-July 2020], wave 2 [August 2020-January 2021], and wave 3 [February-April 2021]) stratified by age: A ≤45 years; B 46–60 years; C 61–75; and D >75 years.**

the subgroups aged ≤75 years, which was more marked among males with critical disease (W2 HR 0.57 [95%CI 0.36–0.91] and W3 HR 0.31 [95%CI 0.17–0.56] *vs* W1), ≥3 comorbidities (W2 HR 0.47 [95%CI 0.19–1.13] and W3 HR 0.37 [95%CI 1.37] *vs* W1) and those with <3 comorbidities (W2 HR 0.75 [95%CI 0.50–1.10] and W3 HR 0.45 [95%CI 0.28–0.71] *vs* W1); however, there was no improvement in survival in any of the subgroups aged >75 years.

## 4. Discussion

The mortality rate among the COVID-19 patients hospitalised at our clinical centre in Milan was lower (15.8%) during the third wave of the epidemic (February-April 2021) than during the first (February-July 2020: 21.3%) or second wave (August 2020-January 2021: 23.7%). It is

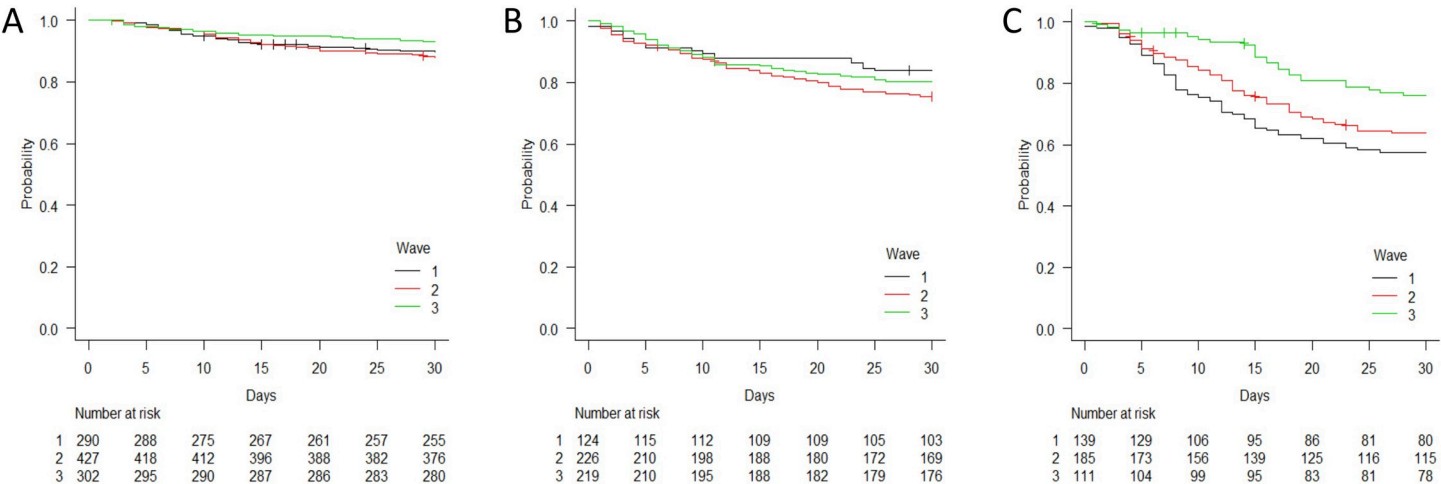

**Fig 3. Time-dependent survival probability by period of hospital admission (wave 1 [February-July 2020], wave 2 [August 2020-January 2021], and wave 3 [February-April 2021]) stratified by disease severity upon hospital admission.** A: mild/moderate disease; B: severe disease; C: critical disease.

interesting to note that a more rapid decrease in in-hospital mortality has been observed in some European studies (as early as during the first or second wave), and that this paralleled an increase in the proportion of women and/or younger patients, who are known to be at lower risk of dying [9–13]. However, when the risk of death of our patients admitted during the different waves of the epidemic was adjusted for the known confounders of age, sex, comorbidities and the severity of COVID-19, we found that those hospitalised during W2 or W3 respectively had a 25% and 42% lower risk of dying than those admitted during W1.

Males accounted for 64.3% of our study population and remained over-represented throughout the three waves, which is in line with other national and international findings [23, 24]. Furthermore, our observation that the risk of death was 36% higher among males than among females is in line with the findings of the OpenSAFELY study, one of the largest cohort studies of the factors associated with the risk of COVID-19-related death, which showed that male gender was associated with a 59% excess risk of mortality [24].

The median age of our COVID-19 patients as a whole was 65.7 years, with 62.2% of the patients being aged >60 years. Patients in the age group of 61–75 years were the most represented during W1 and W3, whereas patients aged >75 years were the most represented during W2, which was also characterised by a higher prevalence of patients with multiple comorbidities. It is well known that an older age and a high morbidity burden are closely associated with an increased risk of COVID-19-related death [25], and so it is likely that these factors together were the major drivers of the greater crude mortality observed during W2.

It is worth noting that, despite the recognised importance of implementing integrated care programmes aimed at protecting the most vulnerable against COVID-19 while mitigating the adverse effects of social isolation [26], the age pattern of the patients enrolled in our study underlines the challenges of ensuring the adequacy of such plans during periods of intense disease recrudescence. This has also been highlighted by other single-centre experiences, and national surveillance data show a peak of COVID-19 cases, hospitalisations and deaths in Italian long-term facilities and nursing homes during October-November 2020, corresponding to W2 [27, 28].

Our observation of a significant reduction in the proportion of patients aged >75 years during W3 may indicate an initial effect of the Italian COVID-19 vaccination campaign, which started in late December 2020, was primarily targeted at-risk populations (including the

elderly) and, as of March 2021 [29], had led to 49.4% coverage among people aged >80 years living in Lombardy.

The results of our multivariable analysis of the risk of death in stratified age groups showed that it significantly decreased in patients aged 46–60 and 61–75 years between W1 and W3, but not in those aged >75 years. A similar figure was observed when patients were stratified into subgroups according to age and other relevant clinical characteristics (sex, disease severity and comorbidity burden). This suggests that the natural history of COVID-19 in elderly patients is not affected by the current means of intervention and, once again, underlines the pivotal role of preventive measures and vaccination in this population [30].

In line with the findings of two previous Italian studies [31, 32], we also observed a trend towards a reduction in the proportion of critically ill patients upon admission between W1 and W3 (from 25.1% to 22.1% and then 17.6%). This change may have been due to increased access to COVID-19 testing and the detection of cases generally associated with a lower risk of death (testing was initially restricted to the most severely ill patients). The results of our multivariable analysis of the risk of death by disease severity upon admission showed a significant decrease in the risk among the patients admitted with critical disease during W2 (a 38% lower risk) and W3 (a 56% lower risk), which may have been due to rapid improvements in respiratory support and pharmacological management.

The proportion of patients who eventually required invasive mechanical ventilation during hospitalisation was highest during W2, but it must be acknowledged that severe resource constraints during W1 may have prevented a number of patients (particularly frailer patients with severe comorbidities) from gaining access to invasive mechanical ventilation [33]. Moreover, the smaller number of SARS-CoV-2 infections recorded in the province of Milan during W3 [22] may have led to a lower proportion of patients with the most critical disease phenotype requiring intensive care.

As noted by others, the significant change in COVID-19 treatment following the first wave of the epidemic reflected an increasing understanding of its underlying pathogenetic mechanisms (including coagulopathy and hyper-inflammation) and the evidence coming from clinical trials. The use of more effective pharmacological strategies such as thrombo-prophylaxis [34] and the appropriate use of corticosteroids [35] and remdesivir [36], combined with an overall improvement in respiratory management [37], may therefore explain much of the reduction in the risk of death observed after W1. However, our finding of a further reduction between W2 and W3 is more difficult to interpret because there were no significant changes in patient management between the two periods, and the cause may lie outside the investigated potential confounders. The results of our sensitivity analysis of the possible effect of the intensity of the wave on the risk of in-hospital death suggest that the reduction during W3 may have been at least partially driven by the lower incidence of COVID-19 in Lombardy [22].

It is unlikely that changes in the virulence of SARS-CoV-2 played a role in the decreasing mortality rate over time because, during W2 and W3, there was an increase in the circulation of the alpha variant (B.1.1.7) in Italy [38, 39], and some studies in the UK have shown that this variant is associated with an increased risk of needing critical care and greater mortality than non-B.1.1.7 viral strains [40]. It can also be excluded that COVID-19 vaccination contributed to the decreased mortality observed during W3 because, although W3 overlapped with the early phase of the Italian vaccination campaign, none of our patients was fully vaccinated against COVID-19, and only two had received one dose of vaccine. On the other hand, it is still important to continue monitoring the case fatality rates associated with COVID-19 and the impact that emerging viral variants and vaccination coverage have on the risk of hospitalisation and COVID-19-related death.

This study has a number of limitations. Firstly, it was a single-centre study so its findings may not be generalisable to other centres in Italy or other countries. However, we think that it is important to continue to collect information from various clinical contexts in order to improve our understanding of how in-hospital mortality evolves in different health systems and patient populations. Secondly, the criteria we used to define the different waves of the epidemic was arbitrary and simply based on the shape of the surges and declines in the number of new COVID-19 cases recorded in Italy during the study period; recently proposed mathematical methods of partitioning time series may better define waves and mortality rates during waves [41]. Thirdly, our dataset did not include data concerning the patients' occupations, socio-economic status, household income, or residence in collective settings such as nursing homes, all of which may contribute to making some groups particularly susceptible to worst health outcomes. Fourthly, although we tried to adjust for the number of SARS-CoV-2 diagnoses in the province of Milan, it is possible that other unmeasured confounders have been missed and that the patients hospitalised during the three waves did not have the same *a priori* probability of death regardless of medical interventions. Finally, a number of structural and human resource-related changes were made during the study period as the pandemic evolved, and these may have affected our findings by introducing biases that could not be accounted for in the analyses.

## 5. Conclusions

The second and third waves of the COVID-19 epidemic were respectively associated with a 25% and 42% lower risk of death than the first wave, and this is probably related to an improvement in the overall management of COVID-19 patients. The patients who received the greatest benefit were those aged 46–60 years and those who were critically ill upon hospital admission, whereas in-hospital mortality among the patients aged >75 years remained unchanged throughout the study period. The dynamic nature of the SARS-CoV-2 pandemic, with the evolution of new variants of concern and their possible effects on the success of the vaccination campaign, suggests the need for further evaluations of the management of COVID-19 patients, the most vulnerable of whom have not changed since the virus first arrived in Italy.

## Supporting information

**S1 Fig. A graphical representation of the number of monthly admissions for COVID-19 during the study period (red bars) and the monthly number of new SARS-CoV-2 diagnoses in the province of Milan (black line).**
(TIF)

**S2 Fig. Time-dependent survival probability in different subgroups by period of hospital admission (wave 1 [February-July 2020], wave 2 [August 2020-January 2021], and wave 3 [February-April 2021]).** 1) A,B,C,D being male or female and aged >75 or ≤75 years; 2) A,B, C,D being male or female, aged >75 or ≤75 years, and presenting with critical disease upon hospital admission; 3) A,B,C,D being male or female, aged >75 or ≤75 years, and presenting without critical disease upon hospital admission; 4) A,B,C,D being male or female, aged >75 or ≤75 years, and with ≥3 co-morbidities; 5) A,B,C,D being male or female, aged >75 or ≤75 years, and presenting with <3 co-morbidities.
(PDF)

**S1 Table. Cox model of the factors associated with death during the second and third waves of the epidemic waves, adjusted for the monthly number of new SARS-CoV-2 diagnoses in the province of Milan.**
(DOCX)

**S1 Dataset.**
(TXT)

## Acknowledgments

The authors would like to thank all of the patients who agreed to participate in this study, and all of the healthcare professionals involved in the care of COVID-19 patients at our hospital.

## Author Contributions

**Conceptualization:** Andrea Giacomelli, Anna Lisa Ridolfo.

**Data curation:** Andrea Giacomelli, Laura Pezzati, Letizia Oreni, Giorgia Carrozzo, Martina Beltrami, Andrea Poloni, Beatrice Caloni, Samuel Lazzarin, Martina Colombo, Giacomo Pozza, Simone Pagano, Stefania Caronni, Chiara Fusetti, Martina Gerbi, Francesco Petri, Fabio Borgonovo, Fabiana D'Aloia, Cristina Negri.

**Formal analysis:** Letizia Oreni.

**Investigation:** Andrea Giacomelli, Laura Pezzati, Giorgia Carrozzo, Martina Beltrami, Andrea Poloni, Beatrice Caloni, Samuel Lazzarin, Martina Colombo, Giacomo Pozza, Simone Pagano, Stefania Caronni, Chiara Fusetti, Martina Gerbi, Francesco Petri, Fabio Borgonovo, Fabiana D'Aloia, Cristina Negri.

**Methodology:** Andrea Giacomelli, Anna Lisa Ridolfo, Letizia Oreni.

**Resources:** Giuliano Rizzardini, Spinello Antinori.

**Supervision:** Anna Lisa Ridolfo, Giuliano Rizzardini, Spinello Antinori.

**Writing – original draft:** Andrea Giacomelli, Anna Lisa Ridolfo.

**Writing – review & editing:** Andrea Giacomelli, Anna Lisa Ridolfo, Laura Pezzati, Letizia Oreni, Giorgia Carrozzo, Martina Beltrami, Andrea Poloni, Beatrice Caloni, Samuel Lazzarin, Martina Colombo, Giacomo Pozza, Simone Pagano, Stefania Caronni, Chiara Fusetti, Martina Gerbi, Francesco Petri, Fabio Borgonovo, Fabiana D'Aloia, Cristina Negri, Giuliano Rizzardini, Spinello Antinori.

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
