## [Decision Letter · Decision Letter 0]

21 Feb 2022

PONE-D-22-01769Mortality rates among COVID-19 patients hospitalised during the first three waves of the epidemic in Milan, Italy: a prospective observational studyPLOS ONE

Dear Dr. Giacomelli,

Thank you for submitting your manuscript to PLOS ONE. After careful consideration, we feel that it has merit but does not fully meet PLOS ONE’s publication criteria as it currently stands. Therefore, we invite you to submit a revised version of the manuscript that addresses the points raised during the review process.

We look forward to receiving your revised manuscript.

Kind regards,

Chiara Lazzeri

Academic Editor

PLOS ONE

Journal Requirements:

Reviewers' comments:

Reviewer's Responses to Questions

**Comments to the Author**

1. Is the manuscript technically sound, and do the data support the conclusions?

Reviewer #1: Yes

Reviewer #2: Yes

2. Has the statistical analysis been performed appropriately and rigorously? 

Reviewer #1: Yes

Reviewer #2: No

3. Have the authors made all data underlying the findings in their manuscript fully available?

Reviewer #1: Yes

Reviewer #2: Yes

4. Is the manuscript presented in an intelligible fashion and written in standard English?

Reviewer #1: Yes

Reviewer #2: Yes

5. Review Comments to the Author

Reviewer #1: This paper is very well written indeed. Almost no typos at all can be found. The paper is pleasant and easy to read. The paper has a robust discussion section, including appropriate limitations. The results are carefully presented and the statistical analysis is sound.

The one limitation that isn’t stated is that any choice of “waves” to divide the pandemic into is somewhat arbitrary. There are more mathematical ways to determine such waves rather than simply by inspection, as proposed in this paper “COVID-19 second wave mortality in Europe and the United States” (https://pubmed.ncbi.nlm.nih.gov/33810707/). It could probably be cited regardless as its subject matter is very similar to yours (reduction of mortality in subsequent Covid waves).

Apart from that, there are some truly minor typos. They may have been introduced in my version of MS Word, perhaps.

co-morbidities->comorbidities

up-dated->updated

re-organisation->reorganisation

Reviewer #2: Manuscript N° PONE-D-22-01769 “Mortality rates among COVID 19 patients hospitalized during the first three waves of the epidemic in Milan, Italy: a prospective observational study”

This is an interesting study that analyzes relevant issues in the history of SARS-CoV-2 infection and COVID-19 disease in a specific and highly relevant Italian hospital. It assessed if and how COVID-19 mortality differed between three consecutive periods in which the knowledge and consequently the management of the disease improved considerably.

However, I see a few major methodological issues and other points in the manuscript that need to be addressed in my opinion. They are as follows:

- I strongly recommend the authors to dismiss statistical significance/null hypothesis testing in the entire study, including data analysis and interpretation. They should instead focus on risk/effect estimates such as HR and AHR together with measures of statistical precision (95% CI) to report and interpret the results, instead of P-values. The black and white approach inherent in statistical significance testing is strongly discouraged nowadays, being methodologically flawed though still largely used (see on this for instance PubMed PMIDs 18582619, 33865412, 28938712, 27209009, 27272951, 28938715, 29650628, 30230362, 30894741, 34453632 and the ASA statement doi.org/10.1080/00031305.2016.1154108). The authors could dismiss all P-values through the manuscript, since they tend always to be interpreted according to the conventional cutpoints of 0.05/0.001. They should instead focus on the size of the effects, dose-response or trend analysis, and stability of the estimates as assessed through the CI width (not just inclusion in it of the unit!). This may have major effect also in the interpretation of the results reported in the abstract: for instance, the authors claim that mortality in W2 – 23.7% - was (‘significantly’) higher than in W1 (21.3%) and W3 (15.8), but actually the absolute difference between W2 and W1 was small, and much smaller than the one between W2 and W3, and W1 and W3. In addition, again in the abstract the authors claim that there was no difference in mortality rates in patients >75 years, but they provide no figures for such a statement, that I suspect was made based on statistical significance testing/P-value cutpoints.

- In the Abstract, please provide the list of relevant confounders that were adjusted for in the analysis, and that allowed to assess W2 and W3 mortality as being ‘significantly lower’ than in W1.

- I see a risk of overadjustment in the large number of variables used to adjust mortality rates in the multivariable model comparing period-specific (W1-W2-W3) mortality. May the authors try also ‘intermediate’ or ‘progressive’ models in terms of adjustment factors, possibly adding the most relevant of them one by one? In addition and more importantly, the authors should carry out stratified analysis, for instance in specific subgroups with reference to sex, age groups, co-morbidities, clinical conditions at hospital admission.. Stratified analyses are always more powerful and informative than multivariable ‘adjusted’ analyses, allowing to identify which subgroups (of hospitalized patients in the present case) experienced different outcomes, independently from the fact that changes in specific subgroups was able to influence the overall estimates.

- The authors should elaborate much more on the potential causes of the decrease over time in the case-fatality rates of hospitalized patients. May also viral variants have played a role?

- The authors should better clarify if they use mortality rate and case-fatality rate is synonymous in the manuscript.

- Tables and Figures: the authors should report mortality rates of hospitalized patients by age groups and sex in the three waves, possibly by plotting them in figures, and also stratifying if possible by comorbidities and other variables such as critical COVID-19 at admission. This in order to clarify which factor(s) drove the mortality decrease over time, including a possible effect of COVID-19 therapy (as correctly hypothesized by the authors in the Discussion) and of vaccination in W3.

- In the Introduction, I strongly suggest the authors to assess more in-depth the literature, focusing on studies that in Italy and outside this country assessed mortality/case-fatality rate of COVID-19, inside and outside hospitals.

- May the authors take into consideration vaccination status in their analyses about changes over time of COVID-19 mortality in hospitalized patients?

6. PLOS authors have the option to publish the peer review history of their article (what does this mean?). If published, this will include your full peer review and any attached files.

Reviewer #1: No

Reviewer #2: No

---

## [Author Response · Author response to Decision Letter 0]

11 Mar 2022

Replies to the Reviewers

Comments from Reviewer 1 : 

We thanks the reviewer for his/her time and suggestions.

Comment #1: The one limitation that isn’t stated is that any choice of “waves” to divide the pandemic into is somewhat arbitrary. There are more mathematical ways to determine such waves rather than simply by inspection, as proposed in this paper “COVID-19 second wave mortality in Europe and the United States” (https://pubmed.ncbi.nlm.nih.gov/33810707/). It could probably be cited regardless as its subject matter is very similar to yours (reduction of mortality in subsequent Covid waves).

Replay:We fully agree with the Reviewer’s comment, and accordingly we have added among limitations of our study (end of the discussion section the following sentence: ‘the criteria that we have adopted for defining waves were based simply on the inspection of the shape of the Italian COVID-19 epidemic trajectory, but should be noticed that mathematical methods in time series analysis have been recently proposed to partition time series in order to better define waves and CFR during waves [Chaos. 2021;31(3):031105.]. 

Comment #2: There are some truly minor typos: co-morbidities->comorbidities;

up-dated->updated; re-organisation->reorganisation

Replay: We have corrected the typos in the text.

Comments from Reviewer 2 : 

General comment This is an interesting study that analyzes relevant issues in the history of SARS-CoV-2 infection and COVID-19 disease in a specific and highly relevant Italian hospital. It assessed if and how COVID-19 mortality differed between three consecutive periods in which the knowledge and consequently the management of the disease improved considerably. However, I see a few major methodological issues and other points in the manuscript that need to be addressed in my opinion.

Replay: We thanks the reviewer for his/her time and constructive feed-back.

Comment #1: I strongly recommend the authors to dismiss statistical significance/null hypothesis testing in the entire study, including data analysis and interpretation. They should instead focus on risk/effect estimates such as HR and AHR together with measures of statistical precision (95% CI) to report and interpret the results, instead of P-values. The black and white approach inherent in statistical significance testing is strongly discouraged nowadays, being methodologically flawed though still largely used (see on this for instance PubMed PMIDs 18582619, 33865412, 28938712, 27209009, 27272951, 28938715, 29650628, 30230362, 30894741, 34453632 and the ASA statement doi.org/10.1080/00031305.2016.1154108). The authors could dismiss all P-values through the manuscript, since they tend always to be interpreted according to the conventional cutpoints of 0.05/0.001. They should instead focus on the size of the effects, dose-response or trend analysis, and stability of the estimates as assessed through the CI width (not just inclusion in it of the unit!). This may have major effect also in the interpretation of the results reported in the abstract: for instance, the authors claim that mortality in W2 – 23.7% - was (‘significantly’) higher than in W1 (21.3%) and W3 (15.8), but actually the absolute difference between W2 and W1 was small, and much smaller than the one between W2 and W3, and W1 and W3. In addition, again in the abstract the authors claim that there was no difference in mortality rates in patients >75 years, but they provide no figures for such a statement, that I suspect was made based on statistical significance testing/P-value cutpoints.

Replay: We thank the reviewer for the observation and we feel to agree with his/her comments. Accordingly, we have modified the text by presenting and interpreting the significance of our data only using the risk/effect estimates and confidence intervals. 

Comment #2: In the Abstract, please provide the list of relevant confounders that were adjusted for in the analysis, and that allowed to assess W2 and W3 mortality as being ‘significantly lower’ than in W1.

Replay: We have added in the abstract the list of all of the relevant confounders we used to adjust our Cox models.

Comment #3: I see a risk of overadjustment in the large number of variables used to adjust mortality rates in the multivariable model comparing period-specific (W1-W2-W3) mortality. May the authors try also ‘intermediate’ or ‘progressive’ models in terms of adjustment factors, possibly adding the most relevant of them one by one?

Replay: We thank the reviewer for his/her suggestion. Attached below you can find the results of a model we have built by introducing one by one variable to an initial model in which we included only age, sex and obesity which could be considered as truly confounders in the casual path of the study. In subsequent models we have progressively introduced variables that could be considered as mediator of the effect, to reach a final model as that presented in the original version of our manuscript. It should be noticed that the estimates obtained for the different waves are consistent in each model with that provided by the final model which was previously presented, so we have preferred to maintain the latter also in view of the fact that all examined variables are commonly evaluated in clinical practice when evaluating the clinical conditions and prognosis of COVID-19 patients. We have not included in the revised version of our manuscript the results of the new analysis, but if the Editor/Reviewer considers it relevant for our data interpretation we could add it as supplementary material.

 AHR AHR Lower bound (95%) AHR Upper bound (95%)

Age (for 10 year more) 1,952 1,801 2,116

Wave 2 vs. 1 0,748 0,594 0,942

Wave 3 vs. 1 0,543 0,415 0,710

Male vs. female 1,659 1,345 2,046

Obesity yes vs. no 1,758 1,411 2,191

 AHR AHR Lower bound (95%) AHR Upper bound (95%)

Age (for 10 year more) 2,156 1,975 2,354

Wave 2 vs. 1 0,736 0,584 0,928

Wave 3 vs. 1 0,562 0,429 0,736

Male vs. female 1,360 1,101 1,679

Obesity yes vs. no 1,641 1,317 2,046

Disease severe vs. mild/moderate 2,119 1,652 2,717

Disease critical vs. mild/moderate 5,019 3,915 6,434

 AHR AHR Lower bound (95%) AHR Upper bound (95%)

Age (for 10 year more) 2,184 1,997 2,388

Wave 2 vs. 1 0,747 0,589 0,946

Wave 3 vs. 1 0,583 0,444 0,767

Male vs. female 1,357 1,096 1,681

Obesity yes vs. no 1,656 1,325 2,070

Disease severe vs. mild/moderate 2,196 1,705 2,828

Disease critical vs. mild/moderate 5,029 3,899 6,486

Time from symptoms onset (x 1 day more) 0,995 0,979 1,011

 AHR AHR Lower bound (95%) AHR Upper bound (95%)

Age (for 10 year more) 2,183 1,993 2,391

Wave 2 vs. 1 0,747 0,589 0,946

Wave 3 vs. 1 0,583 0,444 0,767

Male vs. female 1,357 1,095 1,682

Obesity yes vs. no 1,656 1,323 2,072

Disease severe vs. mild/moderate 2,196 1,705 2,829

Disease critical vs. mild/moderate 5,030 3,898 6,491

Time from symptoms onset (x 1 day more) 0,995 0,979 1,011

Num. comorb. ≥ 3 vs. < 3 1,004 0,788 1,280

Comment #4: In addition and more importantly, the authors should carry out stratified analysis, for instance in specific subgroups with reference to sex, age groups, co-morbidities, clinical conditions at hospital admission. Stratified analyses are always more powerful and informative than multivariable ‘adjusted’ analyses, allowing to identify which subgroups (of hospitalized patients in the present case) experienced different outcomes, independently from the fact that changes in specific subgroups was able to influence the overall estimates.

Replay: We thank the reviewer for the suggested data analysis. We have provided an additional Figure (S2) in which we have plotted the estimated survival probability of specific patients’ subgroups according to the period of their admission (W1, W2, and W3). The subgroups were defined by multiple clinically relevant variables including: sex, age (>75 vs ≤75 years), disease severity (critical vs non-critical) and comorbidity burden (≥3 comorbidities vs <3 comorbidities). Findings were described in the Results section as follows: ‘Results of survival analysis according to the period of admission of subgroups of patients stratified by multiple characteristics are presented in the Supplementary Figure 2. Overall, an improvement in survival was observed in subgroups of patients aged ≤75 years, which was more evident for males with critical disease or with multiple comorbidities. Conversely, there was no significant change in survival in all groups of patients aged> 75 years over the three epidemic waves.

Comment #5: The authors should elaborate much more on the potential causes of the decrease over time in the case-fatality rates of hospitalized patients. May also viral variants have played a role?

Replay: We fully agree with the Reviewer’s comment. At this regard, we have added the following statement in the the discussion section: ‘It is unlikely that changes in SARS-CoV-2 virulence may have played a role in the over time decrease of mortality rate we observed among our cohort of hospitalized patients. In fact, over W2 and W3 an increase in the circulation of SARS-CoV-2 variant alpha (B.1.1.7) has been observed in Italy [Virol J. 2021 Aug 14;18(1):168.], and some studies in UK have shown that this variant is associated with increased risk of critical care need and mortality compared to non-B.1.1.7 viral strains [Lancet Infect Dis. 2021 Nov;21(11):1518-1528]. We have also commented further on this issue as follows: ‘On the other hand, as the pandemic progresses, it is important continuing to monitor case fatality rate associated with COVID-19 and the impact that emerging viral variants and vaccination coverage have on the risk of hospitalization and COVID-19 death’. 

Comment #6: The authors should better clarify if they use mortality rate and case-fatality rate is synonymous in the manuscript.

Replay: In our study conducted on a well defined population of patients hospitalized with COVID-19 the terms ‘mortality rate’ and ‘case-fatality rate’ can be considered synonymous. 

Comment #7: Tables and Figures: the authors should report mortality rates of hospitalized patients by age groups and sex in the three waves, possibly by plotting them in figures, and also stratifying if possible by comorbidities and other variables such as critical COVID-19 at admission. This in order to clarify which factor(s) drove the mortality decrease over time, including a possible effect of COVID-19 therapy (as correctly hypothesized by the authors in the Discussion) and of vaccination in W3

Replay: We thank the Reviewer for his/her suggestion. a) We have provided a new Figure (S2) with the survival analysis for different subgroups stratified by multiple clinically relevant variables (see also response to comment 4). b) We do agree that the effect of different treatments could have impacted toward an improvement of outcome especially for those patients with a critical disease in which an improved knowledge of the disease’s natural history was maturated overtime. Nevertheless, the present study does not allow to draw any inference regarding the effect of different treatments apart from the modification of the standard of care from the 1st wave (oxygen supply plus remdesivir) to that of the 2nd and 3rd waves (oxygen supply plus remdesivir plus dexamethasone plus enoxaparin).

As regards the possible role of vaccination see our response to comment 9. 

- Comment #8: In the Introduction, I strongly suggest the authors to assess more in-depth the literature, focusing on studies that in Italy and outside this country assessed mortality/case-fatality rate of COVID-19, inside and outside hospitals.

Replay: This is a correct observation and accordingly we have focused more on available data regarding overall CFR of COVID-19 reported in Italy and in other countries hit by the pandemic. We have added the following paragraph in the Introduction section:‘Initial analyses performed in April 2020 of data from 9 countries (China, France, Germany, Italy, the Netherlands, South Korea, Spain, Switzerland and Unites States) showed wide variation of the overall case fatality rate (CFR): from 0.7% in Germany to 9.3% in Italy, with two-thirds of the CRF variability being explained by differences in the age distribution of the cases [Ann Intern Med. 2020 Nov 3;173(9):714-720.]. In Italy, the CFR increased to 14% by June 2020, mostly attributable to increasing age-specific case-fatality [PLoS One. 2020 Sep 10;15(9):e0238904].’

- Comment #9: May the authors take into consideration vaccination status in their analyses about changes over time of COVID-19 mortality in hospitalized patients?

Replay: We might exclude a contributing role of COVID-19 vaccination in the decreased mortality rate of hospitalized COVID-19 patients we observed during W3. In fact, although W3 overalapped with the early phase of the Italian vaccination campaign, no patients in our cohort were fully vaccinated against COVID-19 and only two had received one dose of vaccine. We have added this statement in the Conclusion section.

---

## [Editor Report · Decision Letter 1]

30 Mar 2022

Mortality rates among COVID-19 patients hospitalised during the first three waves of the epidemic in Milan, Italy: a prospective observational study

PONE-D-22-01769R1

Dear Dr. Giacomelli,

We’re pleased to inform you that your manuscript has been judged scientifically suitable for publication and will be formally accepted for publication once it meets all outstanding technical requirements.

Kind regards,

Chiara Lazzeri

Academic Editor

PLOS ONE
---

## [Editor Report · Acceptance letter]

1 Apr 2022

PONE-D-22-01769R1 

Mortality rates among COVID-19 patients hospitalised during the first three waves of the epidemic in Milan, Italy: a prospective observational study 

Dear Dr. Giacomelli:

I'm pleased to inform you that your manuscript has been deemed suitable for publication in PLOS ONE. Congratulations! Your manuscript is now with our production department. 

Kind regards, 

on behalf of

Dr. Chiara Lazzeri 

Academic Editor

PLOS ONE